# Systematic Analysis of Combined Antioxidant and Membrane-Stabilizing Properties of Several *Lamiaceae* Family Kazakhstani Plants for Potential Production of Tea Beverages

**DOI:** 10.3390/plants10040666

**Published:** 2021-03-30

**Authors:** Alibek Ydyrys, Nazgul Zhaparkulova, Arailym Aralbaeva, Aigul Mamataeva, Ainur Seilkhan, Sayagul Syraiyl, Maіra Murzakhmetova

**Affiliations:** 1Biomedical Research Centre, Al-Farabi Kazakh National University, al-Farabi Av. 71, Almaty 050040, Kazakhstan; 2Faculty of Biology and Biotechnology, Al-Farabi Kazakh National University, al-Farabi Av. 71, Almaty 050040, Kazakhstan; Nazgul.zhaparkulova@kaznu.kz (N.Z.); Sayagul.Syraiyl@kaznu.kz (S.S.); Maira.Murzakhmetova@kaznu.kz (M.M.); 3Faculty of Medicine and Health Care, Al-Farabi Kazakh National University, al-Farabi Av. 71, Almaty 050040, Kazakhstan; aray3005@mail.ru; 4Department of Food Biotechnology, Almaty Technological University, Furkat Str. 348/4, Almaty 050008, Kazakhstan; mamataevabbm1976@mail.ru; 5Faculty of Natural Sciences and Geography, Abai Каzakh National Pedagogical University, Dostuk 13, Almaty 050020, Kazakhstan; ainura_seilkhan@mail.ru

**Keywords:** antioxidants, plant extracts, flavonoids, lipid peroxidation, tea beverages

## Abstract

One of the most important compounds that exhibit a wide range of biological activities with especially strong antioxidant action are plant polyphenols. In the course of the experiment, the dose-dependent effects of polyphenols-rich extracts isolated from the *Lamiaceae* family Kazakhstani plants were studied on the processes of lipid peroxidation and on the degree of erythrocytes hemolysis. The activity of aqueous-ethanolic extracts from dried parts of plants, such as *Origanum vulgare*, *Ziziphora bungeana*, *Dracocephalum integrifolium*, *Mentha piperita*, *Leonurus turkestanicus*, *Thymus serpyllum*, and *Salvia officinalis*, was studied in a Wistar rat model. Lipid peroxidation (LPO) in liver microsomes was assessed by measuring malondialdehyde content in the form of thiobarbituric acid-reacting substances (TBARS). Estimation of osmotic resistance of isolated erythrocytes was evaluated based on hemoglobin absorbance. The amount of total phenolics in the extracts was measured using the Folin-Ciocalteu reagent method. Based on the results, *Thymus serpyllum* extract exhibited a significantly higher antioxidant activity (IC50 = 3.3 ± 0.7) compared to other plant extracts. Accordingly, among the extracts studied, those from *Salvia officinalis*, *Thymus serpyllum*, and *Origanum vulgare* show the most pronounced membrane-stabilizing activity. Antioxidant and antihemolytic properties of green tea and *Origanum vulgare* extract mixtures were similar to that of each individual plant extract. Similar results were obtained when the green tea extract was mixed with *Mentha piperita*, *Ziziphora bungeana*, and *Dracocephalum integrifolium* extracts, indicating no discernible synergistic interaction.

## 1. Introduction

Plants have a large number of important biochemical compounds that are beneficial to cells, tissues, and the whole organism. Biological activities of such compounds depend on their chemical structure. Polyphenols and flavonoids constitute one of the major groups of secondary metabolites that occur in staple food plants, where they perform a wide array of biological functions [1,2]. Despite the differences in their chemical structure and composition, flavonoids are strong natural antioxidants, which neutralize free radicals, prevent from damaging cells and tissues, and consequently prevent occurrence of pathological processes [3].

Anthropogenic pollution of the environment has undesirable effects on human health, and therefore, the adverse influence of environmental factors on health is becoming increasingly important every year [4]. Unfavorable environmental conditions may be the direct cause of human health problems. Almost 12.6 million deaths each year are attributable to unhealthy environments according to the estimates of the World Health Organization (https://www.who.int). Because they are rich in biologically active compounds, medicinal plants are widely used for therapeutic and prophylactic purposes [5,6].

Polyphenolic compounds, including flavonoids, have a wide range of effects on body cells and systems, produced by plants, which are widely found in fruits and vegetables are thought to protect against ultraviolet radiation, e.g., Reference [7], decreasing the risk of carcinogenesis, complication of cardiovascular disorders, diabetes mellitus, etc., which are accompanied by oxidative stress. In addition, polyphenols are complex antioxidants with increasing public health significance, particularly in the areas of nutrition, epidemiology, and primary, secondary, and tertiary prevention [8]. Therapeutic corrective action of polyphenols is based on their ability to bind and neutralize free radicals [9,10,11].

The free radicals are formed in a body during physiological, biochemical processes and play an important role in the transmission of cellular signals, in the synthesis of prostaglandins and cytokines, and in the utilization and renewal of cellular structures [12]. As a result, antioxidant reserves are depleted, and organisms develop oxidative stress [13]. Increase in lipoperoxidation product levels is the main cause of most pathologies [14]. The amount of free radical oxidation does not exceed physiological limits in healthy organisms, owing to normal functioning of the antioxidant protection system, which neutralizes excess quantities of lipoperoxidation products [15].

Tea beverages represent one of the main groups of food products [16]. They can correct some deficiencies in vitamins, microelements, and other essential nutrients due to valuable components from fruits and berries in their composition. Tea beverages may include black or green tea as basic components or consist of herbs only, which possess health-improving properties [17].

The aim of our investigation was to study antioxidant and membrane-protective properties of some medicinal plant extracts from the family *Lamiaceae*, as well as their content of polyphenolic compounds and flavonoids, to find ways to improve composition of beverages and increase the antioxidant properties of green and black tea.

## 2. Materials and Methods

### 2.1. Laboratory Animals

Male Wistar rats (220 ± 30 g) were housed under the standard conditions of light and dark cycle with free access to food and water. Blood and livers were taken as described below. The experiments were coordinated and approved by the local ethical commission of the Kazakh Academy of Nutrition, Almaty, Kazakhstan (control No. 03/145 of 15 September 2015). All experiments were performed according to national and institutional guidelines, which are in line with guidelines for reporting experiments involving animals: the ARRIVE guidelines and the directive 6 July 2010 [18]. Rats were euthanized by cervical dislocation under isoflurane anesthesia. The livers were isolated, washed, and perfused with chilled saline. Tissue was minced and homogenized (1:10 *w*/*v*) in 10 mM potassium phosphate buffer (pH 7.4) containing 1 mM Ethylenediamine tetraacetic acid (EDTA) on ice. The homogenate was centrifuged at 10,000× *g* at 4 °C, for 20 min. The supernatant was further centrifuged at 100,000× *g*, for 60 min, to obtain the microsomal fraction. The pellet (microsomes) was suspended in a buffer containing 10 mM histidine (pH 7.2), 25% (*v*/*v*) glycerol, 0.1 mM EDTA, and 0.2 mM CaCl2, and was stored at −20 °C. The protein content was measured by the Lowry assay using bovine serum albumin as standard [19].

### 2.2. Preparation of Plant Extracts

“Greenfield” trademark tea was chosen due to its popularity on Kazakh markets, as well as the species diversity of products. Along with it, medicinal plants belonging to *Lamiaceae* family (Oregano—*Origanum vulgare*, Ziziphora bungeana—*Ziziphora bungeana*, Dragonhead—*Dracocephalum integrifolium*, Peppermint—*Mentha piperita*, Motherwort—*Leonurus turkestanicus*, Thyme—*Thymus serpyllum*, Sage—*Salvia officinalis*) were used for investigation. Plant materials were purchased from a local pharmacy and identified by Dr. Alibek Ydyrys. Specimens (*Origanum vulgare*—No. Д- 617, *Ziziphora bungeana*—No. 3-1210, *Dracocephalum integrifolium*—No. 3-1382, *Mentha piperita*—No. 3-2596, *Leonurus turkestanicus*—No. П-3458, *Thymus serpyllum*—No. T-4003, *Salvia officinalis*—No. III-5122) were deposited in the Herbarium of Laboratory Plant Biomorphology, Faculty of Biology and Biotechnology, Al-Farabi Kazakh National University, Almaty, Kazakhstan. Tested plants were crushed and powdered with a laboratory mill. One g of crushed and powdered dried parts of the tested plants was extracted with 10 mL of 50% (*v*/*v*) aqueous ethanol at room temperature, for 20 h in the dark, as described previously [20]. The mixture was then centrifuged at 20,000× *g* for 10 min, and the supernatant was dried at 37 °C in a rotary evaporator. Stock solutions of the dried extracts (100 mg) were freshly prepared in 50% ethanol before use in experiments.

### 2.3. Estimation of Total Phenolic and Flavonoid Content

The amount of total phenolics in the extracts was measured using the Folin-Ciocalteu reagent method [15]: 0.5 mL of each extract (l.0 mg/mL) was added into test tubes containing 2.5 mL of 10% Folin-Ciocalteu reagent and 2.0 mL of 2% sodium carbonate solution, and the tubes were thoroughly shaken. The mixture was incubated at 45 °C with intermittent shaking for 15 min. Absorbance was measured at 765 nm using a PD 303 UV-Vis spectrophotometer (Apel, Japan; here and later). Gallic acid (from Sigma−Aldrich, Milan, Italy) was used as standard to obtain a calibration curve (ranging from 0 to 1 mg/mL). The results were expressed in µg gallic acid equivalents (GAE) per mg dry extract.

The total flavonoid contents were determined using a colorimetric assay, with rutin as standard [16]: 0.5 mL of each extract (l.0 mg/mL) was mixed with 2.0 mL of distilled water and 150 µl of 5% sodium nitrate. After 6 min, 150 µl of 10% aluminum chloride and 2.0 mL of 1M sodium hydroxide were added to this mixture and the resulting final solution was left at room temperature for 15 min. Then, absorbance of the mixtures was measured at 510 nm. The results were expressed in µg of rutin equivalents (RE) per mg of dry extract. The calibration curve was prepared in the same manner using 0–1.0 mg/mL of rutoside solutions in methanol.

### 2.4. Estimation of Lipid Peroxidation in Liver Microsomes

Lipid peroxidation (LPO) was assessed by measuring malondialdehyde content in the form of thiobarbituric acid-reacting substances (TBARS) by the method of Ohkawa et al. [18]. Briefly, liver microsomes were preincubated with vehicle or test agents in a buffer containing 50 mM KH_2_PO_4_ (pH 7.2) and 145 mM NaCl at 37 °C under constant stirring for 10 min. The basal and 0.02 mM Fe^2+^/0.5 mM ascorbate-induced microsomal LPO was then determined in a reaction mixture containing 0.9 M sodium acetate buffer (pH 3.5), 0.4% Sodium dodecyl sulphate (SDS), and 20 mM thiobarbituric acid after incubation at 95 °C for 60 min. After cooling to room temperature, the mixture was extracted by n-butanol:pyridine (15:1, *v*/*v*) and centrifuged at 3000× *g* for 5 min. The organic layer was collected, and its absorbance was measured at 532 nm. The Malondialdehyde (MDA) concentration was expressed as nmol of TBARS per mg of protein [19,20].

### 2.5. Isolation of Rat Erythrocytes

Blood was collected from rats by cardiac puncture under isoflurane anesthesia followed by humane euthanasia. It was centrifuged at 1000× *g*, for 10 min, and plasma and white blood cells were removed. Erythrocyte pellets were washed twice with a buffer containing 5 mM Na2HPO4 (pH 7.4) and 150 mM NaCl and were used immediately for osmotic resistance tests [21].

### 2.6. Estimation of Osmotic Resistance of Erythrocytes

Osmotic resistance of erythrocytes (ORE) was measured as described previously [22]. Isolated erythrocytes were preincubated with vehicle or test agents at 37 °C for 10 min and subjected to a hypotonic solution of NaCl (0.4%) at 37 °C for 20 min, followed by centrifugation at 14,000× *g* for 10 min. Hemoglobin absorbance was then measured in the supernatant at 540 nm. The extent of hemolysis was calculated as the percentage of total hemolysis caused by 0.1% Na_2_CO_3_ [23].

### 2.7. Statistical Data Analysis

The results were statistically processed using the GraphPad Prism 5.01 Program (GraphPad Software, USA). Statistical analysis of data was performed by the methods of descriptive and comparative statistics. The results were represented as mean ± standard deviation (SD) of three independent experiments. The relationship between extract concentration and lipid peroxidation was determined along with the hemolysis degree. Pearson correlation coefficient was calculated in accordance with the nonlinear regression equation, and registered changes in the indices were considered reliable for *p* ≤ 0.05 with the T-criterion. For the statistical analysis of data on such indicators as the content of total polyphenols and flavonoids, the IC50 value for lipoperoxidation, and hemolysis, the nonparametric Kruskal–Wallis analysis of variance was used, and the results were considered statistically significant at *p* ≤ 0.05. Statistical analysis of the data obtained in the study of plant extract and tea mixtures was produced with application of the Student’s *t*-test, and registered changes in the indices were considered reliable for *p* ≤ 0.05 with the Fisher test.

## 3. Results

### 3.1. Properties of Plant Extracts

Results of the study of the content of polyphenolic compounds, total flavonoids, and IC50 for plant extracts are presented in Table 1. By their IC50 value, extracts can be placed in the order: thyme < oregano < peppermint < dragonhead < ziziphora, with the IC50 value for the antioxidant properties of motherwort being higher than those of other species included in this study (Table 1).

Based on estimation of membrane-stabilizing properties, that are associated with the IC_50_ values, a conclusion can be made about the most significant membrane-stabilizing effects of oregano and sage. Extract of thyme at a concentration of 200 μg/mL inhibited hemolysis by 50%, whereas IC_50_ value of other extracts could not be assessed in the tested concentration range.

Based on the results of the study of the content of phenolic compounds and flavonoids, plant extracts can be ranked as: thyme > oregano > peppermint > sage > ziziphora > dragonhead > motherwort. The data obtained correlates with the results of studies of antioxidant and membrane stabilizing properties of plant extracts.

### 3.2. Influence of Herbal Extracts of Family Lamiaceae

As represented in Table 2, the oregano and sage extracts possess the best membrane-protective qualities. Plant ethanolic extracts authentically decreased erythrocyte hemolysis in a concentration range from 0 to 200 μg dry substance/mL. In a concentration of 200 μg/mL, fragility of erythrocytes accordingly decreased by up to 33.4% ± 2.0% and 34.6% ± 2.3%. Antihemolytic action of plantain comes out dose-dependent.

Phytoextracts of peppermint plant, dragonhead, and *Ziziphora Bunga* showed an unsignificant influence on erythrocyte hemolysis at a 25 μg/mL concentration—hemolysis level consists of 99.5%, 95.2%, and 94.5%. Nevertheless, in concentrations above 50 μg/mL, there was a strengthening of *Ziziphora Bunga* and dragonhead extracts’ protective effect observed on erythrocyte membranes. Notably, thyme extracts do not change hemolysis level in concentrations of 50, 100, and 200 μg/mL, but for action of extracts in concentrations higher than 25 μg/mL, there was significant enlarging of erythrocyte membrane resistance observed (Table 2). A comparison of green tea and black tea showed that green teas’ membrane-protective effect is better. Analysis of the results of the research showed that not all plant extracts possess a membrane-stabilizing property and reduce the hemolysis of red blood cells; moreover, motherwort had a damaging hemolytic effect on the membranes of red blood cells. Most of the extracts showed a great change in stability of erythrocyte membranes.

Studies of herbal extracts’ influence on peroxidation processes in liver microsomes resulted that thyme herbs possess highly expressive antioxidative properties (Table 3), namely, extracts totally inhibited malondialdehyde generation in concentrations from 5 to 20 μg per 1 mg protein.


During the experiments, the suspension of erythrocytes was preincubated with extracts of medicinal plants and subjected to a hypoosmotic solution of NaCl. Osmotic resistance was assessed by the degree of hemolysis. As can be seen from Table 2, in the concentration range of 0–200 μg/mL, practically all the extracts led to the dose-dependent decrease in cell hemolysis. These data clearly demonstrated that extracts of oregano, thyme, and sage have antihemolytic activity, and they reduced the level of hemolysis by 17%, 24.6%, and 35.7%, respectively. Ziziphora, dragonhead, and motherwort extract showed antihemolytic properties too, reducing hemolysis of erythrocytes by 43–53% at 200 μg/mL concentrations. Similarly, the peppermint extract exhibits hemolytic properties at a low concentration (25 μg/mL), as evident from a slight increase in the level of hemolysis of the red blood cells.



Accordingly, among the extracts studied, those from sage, thyme, and oregano show the most pronounced membrane-stabilizing activity. The extracts of these plants showed a significant anti-hemolitic effect at a concentration of 25 μg/mL, while a similar effect of the remaining extracts was manifested at concentrations above 50 μg/mL.



Study of tea extracts’ effects on osmotic resistance of cell membranes showed that both types of tea significantly decrease erythrocyte hemolysis in hypotonic solution, due to their membrane-protective activities. However, protective properties of black tea were less than those of green tea. In the concentration range of 0–200 μg/mL, the dose-dependent decrease in the level of hemolysis was 54.6% for the green tea extract and 45.8% for the black tea extract.



Moreover, the extracts of all plants used in this study exhibit antioxidant properties, inhibiting the formation of LPO products. Our data show that the extracts of oregano, sage, thyme, dragonhead, ziziphora, and peppermint have significant antioxidant properties. We found that they almost completely inhibit the formation of LPO products at all concentrations tested.



In contrast, the extract of the motherwort showed a prooxidant effect over the whole concentration range, as evident from a higher MDA value relative to control.



As for the antioxidant properties of tea extracts, the green tea significantly inhibited LPO processes at a concentration of 10 μg/mL, while the same concentration of the black tea extract inhibited formation of TBARS by 35%, and 20 μg/mL concentration inhibited LPO level by around 70%. Further increases in extract concentration led to the complete depression of the free radical oxidation capacity.


### 3.3. Antioxidant Properties and Membrane-Stabilizing Properties of Combination Extracts of Plants and Tea

As noted, at a concentration of 20 μg/mL, the content of TBA-active products was reduced by almost 60–90%. The results of the study of extracts’ potentials are presented in Table 4 and Table 5.

Under the combination of black tea with oregano and peppermint (Table 4), no significant increase in antioxidant and membrane-stabilizing properties has been demonstrated in comparison with the individual plant extracts. *Thymus serpyllum* extract exhibited a significantly higher antioxidant activity (IC_50_ = 3.3 ± 0.7) compared to other plant extracts. The IC50 values for individual extracts ranged as follows: *Thymus serpyllum* > *Origanum vulgare* > *Mentha piperita* > *Dracocephalum integrifolium* > *Salvia officinalis* > *Ziziphora bungeana,* at combination with black tea: *Thymus serpyllum* > *Mentha piperita* > *Ziziphora bungeana* > *Salvia officinalis* > *Origanum vulgare* > *Dracocephalum integrifolium,* and at combination with green tea: *Thymus serpyllum* > *Origanum vulgare* > *Salvia officinalis* > *Mentha piperita* > *Ziziphora bungeana* > *Dracocephalum integrifolium*. This assay indicates that the most active extract was from thyme in combination with black tea. In addition, results showed that the extract was more potent than individual extracts and those in combination with green tea.

The membrane-protective effect of the thyme and black tea mixture was greater than that of each individual extract alone. However, no increase in antioxidant potential was observed. The dragonhead and black tea mixture did not lead to significant changes, and IC_50_ stayed at its original level. In contrast to the above-mentioned plants, mixing salvia and ziziphora extracts with black tea extract resulted in increased antioxidant and membrane-stabilizing effects, which can be explained by a synergic interaction between components in these extracts. Results of the similar studies with mixtures of green tea and other plant extracts are presented in Table 5.

Based on the results of the study of the content of polyphenolic compounds and common flavonoids, plant extracts can be ranked as: thyme > oregano > peppermint > sage > ziziphora > dragonhead > motherwort. The data obtained correlates with the results of studies of antioxidant and membrane-stabilizing properties of plant extracts. Results on the study of antioxidant effect in the concentration range of 0–20 μg are presented in Figure 1.

As noted, antioxidant and antihemolytic properties of green tea and oregano extract mixtures were similar to that of each individual plant extract. Similar results were obtained when the green tea extract was mixed with peppermint, ziziphora, and dragonhead extracts, indicating no discernible synergistic interaction. In contrast, when thyme and salvia extracts were mixed with green tea, the mixtures led to a subsequent increase in antioxidant potential and membrane-stabilizing effects, again indicating a synergistic interaction between components in these individual plant extracts.

## 4. Discussion

Data obtained in the study of the effect of plant extracts on the osmotic resistance of erythrocytes and on the processes of lipid peroxidation in microsomal fraction of liver membranes are presented in Table 2 and Table 3.

It is known that raw materials of plant origin have a multitude of substances, and they have beneficial properties for humans [24,25,26]. Green and black teas are both obtained from the leaves of *Camellia sinensis* that are different by their production technologies. A significant number of studies have been devoted to the studies of antioxidant properties of green and black tea [27,28,29].

Medicinal plants contain various groups of phytochemical compounds that are natural sources of antioxidants. The large ratio of physiologically active phenolic and polyphenolic compounds in tea suggests the possibility for the important role of tea in prophylactics of several diseases [30,31]. There is evidence from both in vitro and in vivo studies that tea can help to reduce the risk of cardiovascular diseases, certain forms of cancer, and a number of other chronic diseases [32,33,34,35]. In order to optimize the antioxidant status of green and black tea and, therefore, enhance their additional protective properties, current studies on the use of plants with significant antioxidant and membrane-stabilizing characteristics were performed.

The phytochemical, antioxidant, and bioavailability of extracts of *Origanum vulgare* was investigated [36]. Antioxidant and bioavailability of plants is a reflection of their phytochemical richness. The type of phytochemicals present in a plant affects its antioxidant activity. These phytochemicals include polyphenols, phenolic acids, flavonoids, and alkaloids, among others, which play overlapping roles in plant defense mechanisms, singlet oxygen scavengers, and high-energy radiation absorbers [37]. A large number of medicinal plants contain flavonoids. The ability of flavonoids to modify membrane-dependent processes, such as free radical-induced membrane lipid peroxidation, is associated not only with their structural characteristics, but also with their ability to interact and integrate into the lipid bilayer [38,39].

Other authors, reporting the antioxidant activity of some plant extracts of the family *Lamiaceae*, including *Ziziphora bungeana*, *Dracocephalum integrifolium*, *Mentha piperita*, *Leonurus turkestanicus*, and *Salvia officinalis* attributed their scavenging activity to phenolic and flavonoid contents [40,41,42,43]. Furthermore, Marchioni et al. also described strong antioxidant effects for Lamiaceae edible flowers in comparison and evaluated the aroma profile spontaneously emitted from the seven selected Lamiaceae flowers [44].

Lipids form the basis of the membrane and play a major role in the structural organization and stabilization of biological membranes [45]. We have previously shown that all used plant extracts exhibit antioxidant properties, protecting membranes from lipid peroxidation. There were evidences that hemolysis of erythrocytes is preceded by membrane lipid peroxidation [46]. In the experiments of the scientist Alinkina Ekaterina Sergeevna, the effect of essential oils on the erythrocytes membranes was shown and the highest activity of essential oils and an extract were found, which included phenols: eugenol, carvacrol, thymol, polyphenols, gingerols, and gingerones. Oils containing 1–3% phenol derivatives had a lower N activity. Low activity was shown by oils containing 10–40% mono- and sesquiterpenes with two double bonds in the cycle. The lowest antiradical activity was exhibited by essential oils, which contained no more than 5% a- and y-terpinenes. For essential oils of thyme, oregano, and savory, the presence of a synergistic effect of thymol and carvacrol was revealed. Also, during the study, it was found that essential oils of oregano, when taken regularly in small doses for 6 months, showed the properties of active bio-antioxidants [47].

It is well-known that black tea loses much of its useful qualities during fermentation processes due to oxidation of the biologically active components. When compared with black tea, green tea has more beneficial effects on living organisms. Both types of tea exert dose-dependent antioxidant effects on membranes of hepatocytes in vitro [48]. Comparison of their antioxidant activities showed that the green tea properties are stronger than those of the black tea. It has also been shown that extracts of oregano, thyme, and sage have a protective effect on both erythrocyte membranes and microsomal fractions of hepatocytes.

## 5. Conclusions

Using plant parts for nourishment, beverage preparation, and disease treatment is a heritage coming from ancient cultures and civilizations. Results of the systematic analysis of antioxidant and membrane-stabilizing properties of Kazakhstani medicinal plants from the *Lamiaceae* family joined with common tea beverages demonstrate especially strong synergistic effects of sage, ziziphora, and thyme extracts, which deserve consideration for further study of individual compounds involved in the observed synergy, and might be recommended for potential production of tea beverages.

Comparison of their antioxidant activities showed that the green tea properties are stronger than those of the black tea. It has also been shown that extracts of oregano, thyme, and sage have a protective effect on both erythrocyte membranes and microsomal fractions of hepatocytes.

## Figures and Tables

**Figure 1 plants-10-00666-f001:**
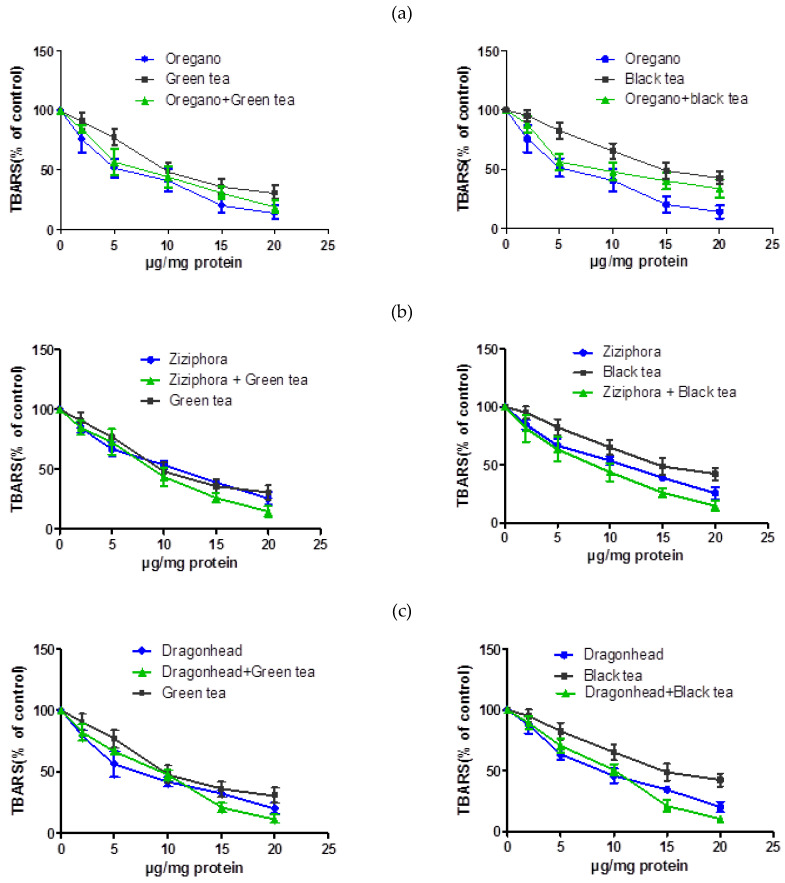
Comparison of single herbal and mixed extracts’ influence on Thiobarbituric Acid Reactive Substances (TBARS) level in liver microsomes in vitro experiments. Note: on the X-axis: extract concentration, μg/mg, along the Y-axis: level of LPO processes, %. (**a**) oregano, (**b**) ziziphora, (**c**) dragonhead, (**d**) thyme, (**e**) sage, (**f**) peppermint.

**Table 1 plants-10-00666-t001:** Lipid peroxidation and membrane-stabilizing properties of several *Lamiaceae* family plant extracts (mean ± standard deviation (SD), n = 3).

Species	Total Polyphenols (μg GAE /mg)	Total Flavonoids (μg RE/mg)	Lipid Peroxidation IC_50_ (µg/mg protein)	Membrane-Stabilizing Properties IC_50_ (µg/mL of RBC)
*Origanum vulgare* *	374.5 ± 15.2	325.2 ± 23.3	5.5 ± 0.9	75.1 ± 8.1
*Ziziphora bungeana* **	401.5 ± 25.6	336.3 ± 42.1	11.5 ± 2.5	194 ± 11.6
*Dracocephalum integrifolium* *	299.4 ± 13.2	157.5 ± 15.3	7.1 ± 1.5	73.5 ± 6.8
*Mentha piperita* ***	137.5 ± 10.2	82.3 ± 7.6	5.8 ± 0.8	>200
*Leonurus turkestanicus* ***	305.2 ± 25.3	285.1 ± 10.2	-	>200
*Thymus serpyllum* *	264.8 ± 9.6	142.3 ± 15.2	3.3 ± 0.7	194 ± 6.5
*Salvia officinalis* *	251.5 ± 16.8	118.2 ± 8.7	9.2 ± 2.8	75.8 ± 4.8
Tea bush—*Camellia sinensis* (green tea)	168.7 ± 8.5	28.7 ± 5.8	9.7 ± 3.1	114.3 ± 9.5
Tea bush—*Camellia sinensis* (black tea)	115.3 ± 8.9	18.8 ± 305	14.8 ± 4.5	>200

The 50% inhibitory concentration (IC_50_) values (μg/mL) were calculated from a log dose concentration-inhibition curve. Total polyphenols and flavonoids concentration are expressed as mean ± SD of triplicate experiments. The significance values for total polyphenols in the Kruskal–Wallace test were *p* = 0.0014 ** (*p* ≤ 0.05), for total flavonoids *p* = 0.0015 ** (*p* ≤ 0.05), for lipid peroxidation IC_50_
*p* = 0.0072 ** (*p* ≤ 0.05), for the sample “*Leonurus turkestanicus*” this indicator was not determined due to the fact that at the studied concentrations it showed a prooxidant effect, for the indicator membrane-stabilizing properties IC50 *p* = 0.0133 * (*p* ≤ 0.05). For the samples “*Mentha piperita*”, “*Leonurus turkestanicus*” and “Camellia sinensis (black tea)”, this indicator was not determined due to the fact that these extracts did not reduce the level of hemolysis by 50% in the studied concentration range.*** *p* ≤ 0.001 Gallic acid equivalent (GAE); Red blood cells (RBC).

**Table 2 plants-10-00666-t002:** Influence of herbal extracts of family *Lamiaceae* on osmotic resistance of erythrocyte membrane. Note: mean ± SD, n = 3. The extent of hemolysis was calculated as the percentage of total hemolysis caused by 0.1% Na_2_CO_3__._

Species	Extract Concentration (μg Dry Substance/mL ES)				
	0	25	50	100	200
*Origanum vulgare* *	100	82.9 ± 3.4	65.7 ± 3.0	40.9 ± 4.8	33.4 ± 2.0
*Ziziphora bungeana* **	100	94.5 ± 4.1	79.8 ± 3.5	66.8 ± 3.6	57.3 ± 6.3
*Dracocephalum integrifolium* *	100	95.2 ± 2.1	89.4 ± 4.5	71.2 ± 4.9	67.3 ± 4.9
*Mentha piperita* ***	100	99.5 ± 2.2	97.9 ± 6.2	89.2 ± 4.5	65.4 ± 3.2
*Leonurus turkestanicus* ***	100	105.6 ± 6.8	101.3 ± 5.6	96.0 ± 5.7	84.9 ± 6.9
*Thymus serpyllum* *	100	74.4 ± 7.6	58.6 ± 3.9	57.0 ± 5.7	50.1 ± 2.9
*Salvia officinalis* *	100	64.3 ± 3.5	61.7 ± 4.9	42.4 ± 8.4	34.6 ± 2.3
Green tea *	100	68.8 ± 5.6	60.2 ± 6.2	52.1 ± 2.3	45.4 ± 2.7
Black tea *	100	87.5 ± 6.5	78.7 ± 6.8	64.1 ± 3.4	53.9 ± 5.5

The data are expressed as the mean ± SD (*n* = 3). Index of Pearson correlation criteria amounted for oregano r_xy_ = −0.9175, for zizifora r_xy_ = −0.9677, for dragonhead r_xy_= –0.9421, for peppermint r_xy_ = −0.9946, for motherwort r_xy_ = −0.9983, for thyme r_xy_ = −0.8856, for sage r_xy_ = −0.9213, for black tea r_xy_ = 0.9499, for green tea r_xy_ = 0.8956. Value of statistical significance of Pearson correlation coefficient amounted for oregano *p* = 0.028 *, for zizifora *p* = 0.007 **, for dragonhead *p* = 0.0166 *, for peppermint *p* = 0.0005 ***, for motherwort *p* < 0.0001 ***, for thyme *p* = 0.0456 *, for sage *p* = 0.0262 * for black tea *p* = 0.0399 *, for green tea *p* = 0.0134 *; Extractable substances (ES).

**Table 3 plants-10-00666-t003:** Effect of herb extracts of family *Lamiaceae* on the level of lipid peroxidation in the liver microsome. Note: mean ± SD, *n* = 3.

Species by Common Name	Extract Concentration (μg Dry Substance/mg Protein)					
	0	2	5	10	15	20
Oregano **	100	77.3 ± 9.6	44.5 ± 4.5	39.0 ± 7.2	20.3 ± 6.5	12.4 ± 2.9
*Ziziphora bungeana* ***	100	84.8 ± 3.9	66.9 ± 5.9	53.9 ± 3.5	39.1 ± 1.6	25.8 ± 5.3
Dragonhead ***	100	91.5 ± 3.2	65.1 ± 4.9	49.4 ± 2.5	34.5 ± 2.9	20.3 ± 4.3
Peppermint **	100	78.7 ± 7.3	62.2 ± 3.6	40.3 ± 4.9	24.4 ± 3.2	16.1 ± 3.9
Motherwort	100	124.3 ± 3.3	117.0 ± 4.8	112.3 ± 7.6	110.1 ± 6.4	106.3 ± 11.3
Thyme *	100	68.4 ± 3.1	26.6 ± 3.7	21.6 ± 4.4	8.8 ± 2.9	4.0 ± 1.3
Sage **	100	84.6 ± 4.4	52.2 ± 5.6	29.3 ± 6.2	14.1 ± 3.7	8.5 ± 2.5
Green tea **	100	90.8 ± 6.7	77.3 ± 7.0	48.2 ± 7.6	35.9 ± 5.9	30.9 ± 6.0
Black tea ***	100	95.1 ± 5.2	82.5 ± 6.8	65.3 ± 6.4	48.8 ± 6.9	42.8 ± 5.4

The data are expressed as the mean ± SD (n = 3), *, *p* < 0.05; **, *p* < 0.01; ***, *p* < 0.001 vs. Control. *, *p* < 0.05; **, *p* < 0.01;*** *p* < 0.001. Index of Pearson correlation criteria amounted for oregano r_xy_ = −0.9316, for zizifora r_xy_ = −0.9811, for dragonhead r_xy_= −0.9794, for peppermint r_xy_ = −0.9698, for motherwort r_xy_ = −0.3653, for thyme r_xy_ = −0.8746, for sage r_xy_ = −0.9508, for black tea r_xy_ = −0.99, for green tea r_xy_ = −0.9722. Value of statistical significance of Pearson correlation coefficient amounted for oregano *p* = 0.0069 *, for zizifora *p* = 0.0005 ***, for dragonhead *p* = 0.0006 ***, for peppermint *p* = 0.0014 **, for motherwort *p* = 0.4765, for thyme *p* = 0.0226 *, for sage *p* = 0.036 **, for black tea *p* = 0.0002 **, for green tea *p* = 0.011 **.

**Table 4 plants-10-00666-t004:** The 50% inhibitory concentration IC_50_ values for the effects of combination extracts of plants and tea (mean + SD).

No.	Sample	IC_50_ (µg/mg Protein, Mean + SD)		
		Individual Extract Mean	In Combination with Black Tea	In Combination with Green Tea
1	*Origanum vulgare*	5.5 ± 0.9	8.9 ± 3.5	6.0 ± 0.8
2	*Thymus serpyllum*	3.3 ± 0.7	2.75 ± 0.4	4.3 ± 1.1
3	*Salvia officinalis*	9.2 ± 2.8	8.2 ± 2.1	7.3 ± 1.6
4	*Mentha piperita*	5.8 ± 0.8	7.3 ± 1.8	8.5 ± 2.3
5	*Ziziphora bungeana,*	11.5 ± 2.5	7.5 ± 0.9	9.1 ± 3.1
6	*Dracocephalum integrifolium*	7.1 ± 1.5	10.3 ± 3.5	9.3 ± 3.8
7	Green tea	9.7 ± 3.1	-	-
8	Black tea	14.8 ± 4.5	-	-

The data are expressed as the mean ± SD (n = 3). The 50% inhibitory concentration (IC_50_) values (μg/mL) were calculated from a log dose concentration inhibition curve.

**Table 5 plants-10-00666-t005:** Membrane-stabilizing properties of combination extracts of plants and tea (mean + SD).

No.	Sample	IC_50_ (µg/mL of RBC, mean + SD)		
		Individual Extract Mean	In Combination with Black Tea	In Combination with Green Tea
1	*Origanum vulgare*	75.1 ± 8.1	113.0 ± 8.5	79.2 ± 6.8
2	*Thymus serpyllum*	194 ± 6.5	178.1 ± 9.8	143.0 ± 7.9
3	*Salvia officinalis*	75.8 ± 4.8	70.0 ± 4.8	65.8 ± 5.6
4	*Mentha piperita*	>200	>200	118.0 ± 12.3
5	*Ziziphora bungeana,*	194 ± 11.6	176.1 ± 8.5	161.0 ± 9.8
6	*Dracocephalum integrifolium*	>200	> 200	173.4 ± 11.5
7	Green tea	114.3 ± 9.5	-	-
8	Black tea	>200	-	-

The data are expressed as the mean ± SD (*n* = 3). The 50% inhibitory concentration (IC_50_) values (μg/mL) were calculated from a log dose concentration inhibition curve.

## Data Availability

Not applicable.

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
