# Peer review of "Systematic Analysis of Combined Antioxidant and Membrane-Stabilizing Properties of Several Lamiaceae Family Kazakhstani Plants for Potential Production of Tea Beverages"

_plants, 2021, doi:10.3390/plants10040666_

Round 1
Reviewer 1 Report
Dear Authors, After reviewing your manuscript I could recommend you to refine the introduction, material and methods, results and discussion (there is missing comparison with the published data).
Author Response
Dear reviewer, we thank you for working on our manuscript, all comments have been taken into account and corrected. In case of any comments, we are ready to work on them.

Reviewer 2 Report
The following sentence is given in the manuscript:
“In order to optimize the antioxidant status of green and black tea and, therefore,
enhance their additional protective properties, current studies on the use of
plants with significant antioxidant and membrane stabilizing characteristics
were performed.”
Explain why an antioxidant action stabilizes the membranes. Mention bibliographical references as to why used plants stabilize membranes. Explain better how the antioxidant state of tea has been optimized.
Author Response
Dear reviewer, we thank you for working on our manuscript, all comments have been taken into account and corrected. In case of any comments, we are ready to work on them.
Answering your questions: Explain why an antioxidant action stabilizes the membranes
To test the antioxidant properties of the studied plant extracts, we used erythrocytes and liver microsomal fraction.
Lipids form the basis of the membrane and play a major role in the structural organization and stabilization of biological membranes.
There is evidence that hemolysis of erythrocytes is preceded by membrane lipid peroxidation:
- Fernandes A., Mira M.L., Azevedo M.S., Manso C. Mechanisms of hemolysis induced by copper // Free Radic. Res. Commun., 1988, 4, â„– 5, Ð .291-298,
- Gallucci T., Lubrano R., Meloni C., Morosetti M., Manca di Villahermosa S., Scoppi P., Palombo G., Castello M.A., Casciani C.U. Red blood cell membrane lipid peroxidation and resistance to erythropoietin therapy in hemodialysis patients // Clin, Nephrol., 1999;52(4):239-245.
- Sivonová , Waczulíková I., Kilanczyk E., Hrnciarová M., Bryszewska M., Klajnert B., Duracková Z. The effect of Pycnogenol on the erythrocyte membrane fluidity // Gen Physiol Biophys. 2004;23(1):39-51.

Reviewer 3 Report
The research made is interesting and the text is well written.My main concerns regarding it are that no hypothesis are stated in the introduction not supported by a state of the art of the litterature on the specific topic.
Then minor complementary data are required in the Material and method section
In the result/discussion it's completely necessary to structure it again there is misunderstanding between the results section (where we can find the discussion) and in the discussion section we can find the result section.
I also recommend to make more detailed statistical analysis to determine which extrat is more effective, in which concentration and which combination.
A more detailed review cn be found directly on the text

Author Response
Dear reviewer, we thank you for working on our manuscript, all comments have been taken into account and corrected. In case of any comments, we are ready to work on them. All your comments have been corrected and sent as a file.
Cover letter
Article
Systematic analysis of combined antioxidant and membrane-stabilizing properties of several Lamiaceae family Kazakhstani plants for potential production of tea beverages
To test the antioxidant properties of the studied plant extracts, we used erythrocytes and liver microsomal fraction.
Lipids form the basis of the membrane and play a major role in the structural organization and stabilization of biological membranes.
There is evidence that hemolysis of erythrocytes is preceded by membrane lipid peroxidation:
- Fernandes A., Mira M.L., Azevedo M.S., Manso C. Mechanisms of hemolysis induced by copper // Free Radic. Res. Commun., 1988, 4, â„– 5, Ð .291-298,
- Gallucci T., Lubrano R., Meloni C., Morosetti M., Manca di Villahermosa S., Scoppi P., Palombo G., Castello M.A., Casciani C.U. Red blood cell membrane lipid peroxidation and resistance to erythropoietin therapy in hemodialysis patients // Clin, Nephrol., 1999;52(4):239-245.
- Sivonová , Waczulíková I., Kilanczyk E., Hrnciarová M., Bryszewska M., Klajnert B., Duracková Z. The effect of Pycnogenol on the erythrocyte membrane fluidity // Gen Physiol Biophys. 2004;23(1):39-51.
Isotonic solution for erythrocytes is 0.85% sodium chloride solution. In 0.48–0.44% NaCl solutions, the least resistant erythrocytes are destroyed (minimal osmotic resistance, upper limit of resistance). Therefore, in a 0.4% NaCl solution, erythrocytes undergo hemolysis, the use of plant extracts increases the resistance of erythrocytes to hemolysis and thus increases the resistance of red blood cells.
The results of our studies suggest that the protection of erythrocyte membranes from hemolysis in a hypotonic NaCl solution occurs as a result of inhibition of membrane phospholipid peroxidation.
In addition, we conducted experiments on the effect of plant extracts on the formation of peroxide products in liver microsomes. Peroxidation in mirosomes was induced 0.02 mM Fe2+/0.5 mM ascorbate. As our experiments showed, all investigated extracts reduced the formation of peroxidation products in liver microsomes.
A large number of medicinal plants contain flavonoids. It is assumed that the ability of flavonoids to modify membrane-dependent processes, such as free radical-induced membrane lipid peroxidation, is associated not only with their structural characteristics, but also with their ability to interact and integrate into the lipid bilayer.
1- Saija A., Scalese M., Lanza M., Marzullo D., Bonina F., Castelli F. Flavonoids as antioxidant agents: importance of their interaction with biomembranes // Free Radic Biol Med, 1995;19(4):481-486. doi: 10.1016/0891-5849(94)00240-k.
2 - de Oliveira NKS, Almeida MRS, Pontes FMM, Barcelos MP, de Paula da Silva CHT, Rosa JMC, Cruz RAS, da Silva Hage-Melim LI. Antioxidant Effect of Flavonoids Present in Euterpe oleracea Martius and Neurodegenerative Diseases: A Literature Review // Cent Nerv Syst Agents Med Chem. 2019;19(2):75-99. doi: 10.2174/1871524919666190502105855.
We have previously shown that all used plant extracts exhibit antioxidant properties, protecting membranes from lipid peroxidation (Araylim N. Aralbaeva, Aigul T. Mamataeva, Nazgul I. Zhaparkulova, Raisa S. Utegalieva, Marina Khanin, Michael Danilenko, Maira K. Murzakhmetova "A composition of medicinal plants with an enhanced ability to suppress microsomal lipid peroxidation and a protective activity against carbon tetrachloride-induced hepatotoxicity").

Round 2
Reviewer 2 Report
The new version of the manuscript takes into account the observations of the referees
Author Response
Dear Reviewer,
Thank you for reviewing our manuscript and accepting our correction based on your comments!
Reviewer 3 Report
There is still a problem of exposition of the result, showing either first the conclusions then the results or the discussion before the results which is still needs to be corrected.
The statistical data should be better mentionned in each table not just answering the reviewers. As weel as the origin of chemical products (Company, Country, city)
I've seen the other modifications in the other parts if convenient for me

Author Response
Dear Reviewer,
We have amended the manuscript based on your comments and suggestions.
1. We have included the resulting sentences in the discussion and conclusion section.
2. Added explanations and processing of statistical data.
3. We have included the country and city of origin of the chemical product gallic acid in the content of the manuscript.
4. We re-checked the content of the manuscript.
Thank you for all your comments and suggestions!
If you have any further comments or suggestions regarding the manuscript, we look forward to hearing from you.
